# Psychosocial impact on frontline health and social care professionals in the UK during the COVID-19 pandemic: a qualitative interview study

Henry Aughterson , Alison R McKinlay, Daisy Fancourt , Alexandra Burton

Department of Behavioural Science and Health, University College London Research Department of Epidemiology and Public Health, London, UK

**Correspondence to**
Henry Aughterson;
henry.aughterson.14@ucl.ac.uk

## ABSTRACT

**Objectives** To explore the psychosocial well-being of health and social care professionals working during the COVID-19 pandemic.

**Design** This was a qualitative study deploying in-depth, individual interviews, which were audio-recorded and transcribed verbatim. Thematic analysis was used for coding.

**Participants** This study involved 25 participants from a range of frontline professions in health and social care.

**Setting** Interviews were conducted over the phone or video call, depending on participant preference.

**Results** From the analysis, we identified 5 overarching themes: communication challenges, work-related stressors, support structures, personal growth and individual resilience. The participants expressed difficulties such as communication challenges and changing work conditions, but also positive factors such as increased team unity at work, and a greater reflection on what matters in life.

**Conclusions** This study provides evidence on the support needs of health and social care professionals amid continued and future disruptions caused by the pandemic. It also elucidates some of the successful strategies (such as mindfulness, hobbies, restricting news intake, virtual socialising activities) deployed by health and social care professionals that can support their resilience and well-being and be used to guide future interventions.

### Strengths and limitations of this study

► This is the first known study to interview both health and social care professionals working across the entirety of the UK, on their experiences working through COVID-19.
► This study used a strong theoretical approach to inform the topic guide, and one-to-one interviews allowed in-depth analysis of the psychosocial experiences of health and social care professionals, complementing the wider availability of quantitative evidence.
► We interviewed a wide range of professions, which provided breadth of experience but might limit the specificity of findings.
► Given the fluctuating nature of the pandemic, attitudes of health and social care professionals may change over time.
► This can be challenging to capture during a single interview, however we did ask questions on how their experience had progressed longitudinally.
► Our sample may have been biased towards people who had more free time to participate and so were coping better than others; however, our sample still described a number of stressful experiences during the pandemic, and it is also possible that workers who were frustrated or stressed wished to express their views.

## INTRODUCTION

To control the spread of the COVID-19 pandemic, epidemiological measures were taken across the globe, with responses differing between nations depending on their own public health circumstances, scientific advice and political priorities.[1] In the UK, from 23 March this involved a national 'lockdown', involving significant restrictions on citizens' way of life including measures such as 'staying at home', social distancing and the closure of workplaces, shops and other services.[2] Specific lockdown measures were eased over time, but major constraints and the progressive tightening and relaxing of such restraints remained for substantial periods.

Some professions, known as 'key workers', considered to provide an essential service to the public, were excluded from various restrictions and continued working throughout the pandemic. Crucially, health and social care professionals were designated as key workers to enable their continued support for patients and clients throughout the UK's National Health Service (NHS) and social care system. When measures were first announced, significant concerns arose around lack of capacity within the NHS, limited personal protective equipment (PPE) and staff burnout.[3]

BMJ

Previous research exploring the psychological impact on health and social care professionals during epidemics such as SARS and Middle East Respiratory Syndrome (MERS), has highlighted the adverse psychological effects that frontline health work during epidemics can have.[4–6] There is also emerging evidence during the COVID-19 pandemic that healthcare workers experienced heightened levels of stress and anxiety,[7–11] depression[8 9 12] and poor sleep quality.[8 13]

There are a number of reasons why health and social care workers can experience adverse psychological consequences in epidemics. First, rising cases of a new infection can lead to longer hours, more intense working environments and work-life imbalance, which disrupt the equilibrium between work demands and workers' response capacity.[14] This, coupled with a lack of control, unclear job expectations and lack of social support at work are the components of 'professional burnout'.[15] Concerns about the mental health and well-being of health and social care professionals in the UK were growing prior to the COVID-19 pandemic, with 'professional burnout' recognised as a particular challenge.[15 16] Moreover, there is evidence one may feel they lack the tools to manage ('loss of manageability') the confusion created by diagnosing and treating an unknown infection ('loss of comprehensibility') and experience a reduction of work to essential rather than meaningful patient interactions ('loss of meaningfulness') which combined may disrupt their 'sense of coherence' (SOC) (a measure of resilience— how effectively one is able to cope with stressors).[6 17–19] This disruption has been found to adversely affect mental health (the SOC theory informed part of our interview guide; see 'Methods' section).[20] However, equally there is evidence demonstrating that health and social care workers have moderate to high levels of psychological resilience during times of pandemics,[21] and so it is unclear whether or not they will have a robust, or disrupted, SOC during COVID-19. Third, staff may be concerned about their own risks from exposure to a new pathogen, or the risks that they might infect family or friends. These concerns can be particularly acute when the aetiology and outcomes from a new virus are not well understood.[6]

There are few published qualitative studies that have investigated the psychosocial impact of the COVID-19 pandemic on both health and social care professionals within the UK. One qualitative study interviewing 30 hospital-based healthcare workers in the UK found heightened anxiety related to PPE issues and lack of training in new skills,[17] while another study with nurses and support workers in care settings identified a lack of pandemic preparedness, heightened anxiety, shortage of PPE and ever-changing PPE guidance.[22] It is unclear what the psychosocial challenges faced by other health and social care workers are—such as general practitioner (GPs), mental health and social workers. Moreover, current research on health and social care workers during COVID-19 has predominantly been quantitative, using pre-assumed hypotheses of negative effects. However,

in previous pandemics such as SARS and MERS, there were some positive outcomes including a more positive outlook towards work, growth under pressure, greater comradery with colleagues and a strong sense of professional responsibility and personal development.[5 6] A UK study interviewing hospital-based healthcare workers during COVID-19 also found increased solidarity between colleagues and high levels of morale.[17]

There is a need for qualitative research to explore factors that have helped to alleviate distress and build resilience among health and social care workers during the pandemic. This is crucial in order to provide richer data of their experiences to aid our understanding of specific stressors, guide future support and interventions both as COVID-19 continues, and also in the occasion of future pandemics and stressful situations within the NHS. There may be common factors that contribute towards individuals' resilience during this period which may provide useful learning for employers and employees in order to better harness or encourage such factors. Therefore, the aims of this study were to explore[1]: the impact of the COVID-19 pandemic overall on the working lives and mental health of health and social care professionals, and[2] the factors that help alleviate distress and contribute to the resilience of health and social care professionals during the pandemic.

## METHODS
### Sample and recruitment
We recruited health and social care professionals from across the UK using social media, personal contacts, newsletters and from a sample of participants taking part in a large, nationwide, quantitative survey study: the UCL COVID-19 Social Study.[23] This research forms a qualitative component of this larger study. Twenty-five participants, from a range of frontline professions within health and social care, were recruited and interviewed between 1 May and 17 September (table 1). Sampling was purposive to include a range of ages, ethnicity, gender and professional roles. Interviews ceased when data sufficiency was reached and the lead author identified no new themes. Presentation of recruitment, data collection and analysis are aligned with the COnsolidated criteria for REporting Qualitative research criteria for reporting qualitative research.[24]

### Data collection:
Semi-structured, one-to-one, telephone or video interviews were conducted by HA (PhD student and trainee medic) and AB (mental health services researcher) exploring the impact of the pandemic on participants' social lives, work life and mental health. The interviews lasted an average of 51 min (range 29–93). Berkman's social networks framework and Antonovsky's SOC theory informed the topic guide questions on social life and mental health, respectively, for example, including questions on social networks, social roles and meaning/purpose that related

**Table 1** Characteristics of the health and social care professionals

| Number of participants | 25 |
|---|---|
| Profession | Hospital doctor (6)<br>General practitioner (4)<br>Hospital nurse (3)<br>Social worker (3)<br>Home carer (2)<br>Care home carer (2)<br>Assistant psychologist (1)<br>Community nurse (1)<br>Practice nurse (1)<br>Counsellor and psychotherapist (1)<br>Academic physiotherapist (1) |
| Age | Range 26–65 years |
| Gender | Male 5<br>Female 20 |
| Ethnicity | White British 17<br>Asian 3<br>Black British 2<br>White and Asian 1<br>White Irish 1<br>White other 1 |

to each theory.[20 25] The full topic guide is provided in online supplemental material and exemplary questions are provided in figure 1. All participants were given a Participant Information Sheet and encouraged to ask questions. Written informed consent was then obtained and a demographics form was completed by all participants. We audio-recorded interviews with participants' consent, and recordings were transcribed by a professional transcription service. The audio recording of interviews and available transcripts enabled repeated revisiting of data in order to remain true to participants' original accounts, helping to enhance validity of the results.

### Patient and public involvement

The study participants or public were not involved in the design of the study, the conduct of the study, the writing of the paper nor in the dissemination of the study results. However, participants will be sent study results if requested, and the findings will be shared with the wider public through newsletters (the MARCH network) and social media.

- How would you describe your social life now that social distancing measures have been brought in because of COVID-19?
- In what ways has your work life been impacted by the COVID-19 pandemic?
- How do you feel about the changes that have been brought about by COVID-19? Have they had any impact on your mental health or well-being?
- Have there been any positive experiences for you resulting from the COVID-19 pandemic?

**Figure 1** Examples of questions in the topic guide

### Data analysis

The analytical approach deployed was reflexive thematic analysis.[26] This followed the steps described by Braun and Clarke[27] of familiarisation with the data, generation and definition of codes, theme searching and producing the report. HA and ARMcK (research psychologist) independently coded four transcripts, which were discussed before HA coded and interpreted all remaining transcripts, continuing until data sufficiency was reached and no new codes identified. A deductive approach was used to develop the initial coding framework based on concepts in the topic guide, followed by an inductive coding approach (where new themes were generated from our data) as new concepts were added to the framework based on the data. Contradictory data and context around codes were retained to capture subtle nuances. Codes were then grouped into themes, with each theme representing a meaningful pattern in the data. All final themes were agreed by study authors. We have included rich and verbatim descriptions of participants' accounts in order to support these findings. Weekly team meetings with researchers from the qualitative COVID-19 Social Study team were also used to discuss and develop findings. This method helped to reduce individual-level research bias that may have affected the interpretation of results, thus enhancing validity of the findings. The software used for coding was NVivo qualitative data analysis, V.12.[28]

### RESULTS

We interviewed 25 participants from a range of professions within health and social care including doctors, nurses, carers and social workers working in hospital, residential, community and primary care settings. Participants were aged 26–65 years, predominantly female (80%) and White British (68%).

### Themes

Five primary themes were identified. These were: communication challenges, work-related stressors, support structures, resilience and personal growth. Themes and corresponding subthemes are displayed in figure 2.

### Communication challenges

The pandemic brought with it numerous challenges around communication for health and social care professionals in their work. One of these was associated with new virtual means of consulting, and the other with the greater levels of difficult conversations.

#### Virtual consulting

Some consultations had shifted online, especially among GPs, therapists and social workers. Several participants said that one of the key benefits was the increased efficiency of virtual communication and consultations.

> Much more is done remotely, which is just so much more efficient, because patients don't always need

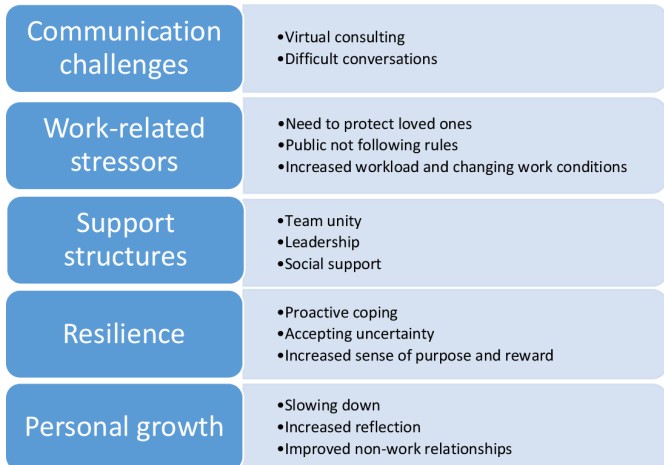

| | |
|---|---|
| **Communication challenges** | • Virtual consulting<br>• Difficult conversations |
| **Work-related stressors** | • Need to protect loved ones<br>• Public not following rules<br>• Increased workload and changing work conditions |
| **Support structures** | • Team unity<br>• Leadership<br>• Social support |
| **Resilience** | • Proactive coping<br>• Accepting uncertainty<br>• Increased sense of purpose and reward |
| **Personal growth** | • Slowing down<br>• Increased reflection<br>• Improved non-work relationships |

**Figure 2** Themes and subthemes.

to be seen face to face. Patients preferred it, we preferred it. (Participant_16_GP)

However, more often participants talked about the limitations of virtual consultations. Participants found it difficult to provide appropriate emotional support, especially in a time where there was heightened need. This was more commonly an issue for social care, mental health and palliative care professionals supporting vulnerable adults or children.

It makes me realise the importance of seeing someone face to face to actually support them. I just don't think a telephone call or a Zoom call is sufficient when it comes to helping people who have profound mental health issues, or even mild mental health issues. And I think that some people just need the power of touch or a hug or a face-to-face human person to ensure that they're kept safe and okay. (Participant_4_Hospital doctor_palliative_care_registrar)

Participants often found it difficult to build new relationships virtually, especially when working with children:

I think if you already know someone well, and you are speaking to them on video call, then that's fine because you've already established the relationship… but if you have to start establishing the relationship on a video call…there is something missing I think, especially working with children… especially if you have a child that's introverted or struggles to communicate or has learning difficulties or is very shy. It's harder to make them feel comfortable when you're on video call. (Participant_10_social_worker)

Participants talked about difficulties identifying crucial signs of deterioration in health from patients or clients, for example, from body language:

We have to be able to pick up signs, for instance, if they are suicidal, I think there's an anxiety there that doing it online, it's difficult sometimes to do that, to pick up on some nuances of the way they

talk… we can't see their whole body language. (Participant_9_counsellor_and_psychotherapist)

Participants, especially mental health and social workers, reported difficulty ensuring a confidential space when consulting virtually:

When they're (children) talking to you, perhaps you see them in a room and it seems like they're alone. But actually, maybe they have all their family members that are standing in the corner. (Participant_10_social_worker)

### Difficult conversations

A common challenge during this period was an increase in difficult conversations with patients, clients and their family members. The need for PPE, and virtual consultations, accentuated these difficulties. This was especially apparent in those working in services that provide mental health and palliative care.

Family communication is awful these days… You can't see someone, and you're speaking to them on the phone, and you're telling them that they can't come and see their loved one, and that their loved one might well die, that's an awful conversation to have with someone…it's definitely one of the worst things about COVID for me. (Participant_8_Hospital_doctor)

Several participants said they were providing more emotional support to patients and clients than usual:

There's been a huge amount of emotional support that we've had to give through anxiety, through grief. All that has been heightened quite greatly really. And a deeper sense of sadness in yourself, that you're trying to support people and having that empathy for them, thinking this is just absolutely horrendous for them. (Participant_20_Home_carer)

### Work-related stressors

Health and social care professionals said that they experienced a range of challenging emotions and psychological difficulties. This included frustration at members of public not following social distancing rules, the concern about protecting their loved ones from infection and increased workload and changing working conditions.

### Public not following the rules

Participants said that they felt frustrated with members of the public not following social distancing and other guidelines, feeling that their work on the frontline was being undermined.

I've been quite annoyed… you're trying your best in lockdown to obey the government guidelines and I think I've had a huge amount of frustration by hearing and seeing people who haven't…certain politicians as well that haven't stuck to guidelines, and I feel sometimes I've been working my socks off and

felt quite cheated. (Participant_20_Home_carer and nurse)

I didn't see my boyfriend for six months. I just did not want to leave my house because of how I feared and felt that other people were behaving… just seeing people not following the rules. (Participant_24_Assistant_psychologist)

### The need to protect loved ones

Nearly all participants spoke about ongoing worry for friends and family compounded by a fear of transmitting the virus to them—due to participants' increased risk of catching it at work. Participants reported taking extra measures in order to protect their loved ones.

Normally I would see my mum every day… but where I work, I was worried, I was more at risk to catch anything, so I definitely didn't. My sisters were actually going into the garden, and talking to her… but I wouldn't, so I was standing at the gate. So that was pretty hard, because I'm very close to my mum. (Participant_21_Care_home_manager_and_carer)

It would be super lovely to have a giant hug from my dad, but I know that's not possible… in the line of work that I do, the risk to him would just be immense because I have been on the COVID wards. (Participant_4_Hospital_doctor_palliative_care_registrar)

### Increased workload and changing work conditions

Fatigue and exhaustion were commonly reported by participants throughout this period.

I'm just feeling really run down… I literally have 4 or 5 weeks where I've not left the house unless it's just to pop to the supermarket… I find that really does impact me… it's like work has taken over my whole life and I'm exhausted. (Participant_11_Family_support_worker)

While not unanimous, some participants said that they were working longer hours and had an increased workload.

My routine was really like… wake up, eat something, go into work, which as shifts as nurses we had to stay in the hospital for 12 and a half hours…go home and eat something, drink something, go to sleep… then wake up and then go to work again… we have been extremely busy compared to the normality. (Participant_19_Hospital_nurse)

Feelings of fatigue were also enhanced by the tenuous nature of PPE:

you just become quite tired…it culminated with masks, visors, aprons, hot weather and regulations changing and sometimes you'd come home from a shift and feel you'd been pulled in all directions really (Participant_20_Home_carer and nurse)

Moreover, a common concern for participants was making decisions that balanced the complex, and often

unknown, risks associated with the virus along with other health risks. This was particularly difficult during the early phases of the pandemic, when less was known about the virus.

…sometimes I've thought, right, we do need to bring patients in, but then are you putting them at more risk exposing them to the virus, which could actually kill them, and actually they could potentially just have a gastro bug and not bowel cancer? (Participant_16_GP)

### Support structures

The availability of support structures at work and home was identified as an important buffer for the psychosocial impact of working during the pandemic and in coping with considerable work-related changes. Themes associated with this feeling of being supported included team unity and leadership at work, and wider social support.

### Team unity

Participants often felt they were closer to their team, and that team unity had increased during the pandemic, united over a common cause. This was more likely if teams were cohesive before the pandemic, and was particularly apparent for doctors in primary and secondary care.

just having the vibes that we're all in this together and we're all going (through) the same thing, and we're pulling in the same direction (Participant_3_Hospital_doctor_intensive_care)

Some of this unity was facilitated through virtual communication:

We actually started up a group, ourselves, on WhatsApp. It's just our team, it doesn't include management. It's just for family support workers and social workers… we try not to put work stuff on there… we try to send each other funny messages or memes… to keep us going. (Participant_11_family_support_worker)

Participants felt the increase in virtual meetings improved attendance, due to the ease of just being able to 'dial in' and improved collaborative working among multidisciplinary teams.

When we're safeguarding a child, you are supposed to work collaboratively… it'll be social care and school, health, maybe a youth organisation or domestic abuse organisation… when you're doing these physical meetings, beforehand, people just wouldn't turn up… but now, they can just dial in. (Participant_23_Social_worker)

However, team unity was not unanimous, with several participants experiencing loneliness, due to an increase in lone working or working from home.

I was then at home isolating for two weeks, and so I was working from home. I had remote access to my computer, so I was doing purely telephone consultations from home and I felt very isolated there, and I didn't feel like I was part of the team at all. (Participant_13_GP)

A few participants also described difficulties connecting with colleagues virtually for support:

I think it could have been improved by seeing each other face-to-face, and I have to say that has been really detrimental to our team. I think you lose a lot by not seeing someone face to face, in terms of their body language and non-verbal cues aren't always captured particularly well through IT. (Participant_4_Hospital_doctor_palliative_care_registrar)

### Leadership

Participants expressed frustration about government handling and changing advice throughout the pandemic. In particular, participants talked about confusing guidance received from management and government regarding PPE or distancing procedures at work, as well as the speed in which the guidance changed.

I know they (upper management in NHS) have difficult decisions to make quickly but I sometimes find their rule making quite vague. A bit like the government, I feel like they're making it up as they go along somewhat. And it changed every day so you'd log onto your emails and there'd be some new change. (Participant_15_community_mental_health_nurse)

However, some felt the culture of blame itself was frustrating:

the other thing that annoys me sometimes is that… everyone wants to blame everybody…it's like the blame game…everybody has to blame Boris… Somebody's responsible. China's responsible… I feel that illnesses and viruses have been around forever and ever and ever… you can't really be pointing the finger all the time, and I find it quite depressing (Participant_12_Hospital_doctor_critical_care)

Most participants felt supported at work and received regular emotional and practical check-ins from management. A small number felt ignored by management, which led to feelings of being overwhelmed:

I was quite anxious about being in the office with COVID I had some colleagues of mine who were able to work from home… I was told that this wasn't possible… it was business as usual. It was a real sense of frustration, not feeling that you're being listened to by my manager and just a sense of feeling overwhelmed and quite helpless about the situation. (Participant_24_Assistant_psychologist)

### Social support

Most participants had supportive relationships with family, friends and colleagues, which helped contribute to a sense of resilience among participants.

I've got a good, strong marriage and we're a good partnership, my husband and I, and we've supported each other through all this… our friends are going through similar things, so I'm able to talk to my friends as well. I've got a good network of friends, so I'm very lucky. (Participant_20_Home_carer nd nurse)

They also felt supported by their local community, and the public, for example, through the 'Clap for Carers' movement where people across the country stood outside their front door once a week to applaud health and social care workers for their contribution:

the Clap for Carers thing, the neighbours would come out and clap, and I found that quite touching… quite uplifting actually (Participant_12_Hospital_doctor_critical_care)

### Resilience

Despite the difficulties and challenges, participants also demonstrated significant psychological resilience. This was often to do with their proactive coping mechanisms, their ability to 'accept uncertainty' and the increased sense of purpose associated with their work.

### Proactive coping

Most participants said they used well-developed coping mechanisms to deal with the ever-changing circumstances, including engaging in hobbies, participating in virtual activities and maintaining routines.

I do genuinely believe getting out into nature has a really good impact on one's mental health and it certainly does on mine…Crafting, doing things, keeping busy…that's really important for your mental health having that occupations. I crochet… we paint, we draw, we make jewellery… (Participant_15_Community_mental_health_nurse)

Participants discussed the negative impact of constant news coverage on coronavirus and the death count, and intentionally restricted their news intake as they felt it was unhelpful, even harmful to their mental health:

the virus messes with your head more than it does your body, if you're not hospitalised. That's just down to the media at the end of the day. There's so much media and so much emphasis on death, not so much on recovery (Participant_7_Hospital_nurse)

### Accepting uncertainty

Most participants had a degree of personal, psychological resilience linked to an acceptance, or 'letting go', of what

they had no control over, such as the overall outcome of the pandemic or government restrictions.

> I'm generally much less anxious now than I was in January and February. And part of that, I think, is about thinking a lot more about death and being a lot more accepting about death, and about what you can control and what's out of your control. (Participant_14_GP)

Part of this philosophy seemed to be influenced by their previous experiences, and their profession:

> Because of my job, I think I'm aware that we're not really in control of lots of things in our life… I see that all the time with patients and people I care for… we don't have control over everything and we have to have a level of acceptance for that. (Participant_15_community_mental_health_nurse)

### Increased sense of purpose and reward

Participants often expressed gratitude for being able to continue working. This brought with it benefits such as purpose, daily structure, predictability, a degree of socialising and being 'in-the-know' about the virus. Participants also talked about how it felt good to be able to contribute. This was highly rewarding and brought a heightened sense of purpose.

> …some people would view me as not being lucky, because I'm a frontline worker, but that's my job and that's what I'm trained to do, so I don't view that as unlucky… I have been lucky in the sense that I've been able to keep busy, to keep working, to feel I'm contributing (Participant_17_GP)

Guilt occurred when participants were not able to contribute as much as they would have liked, for example, they were told they could not work because they were at high risk, or had to work from home.

> I felt incredibly guilty by the fact that I wasn't helping out on the frontline because I was pregnant. I was just told I'm not allowed to see any patients, by occupational health, and sent home. (Participant_6_Hospital_doctor)

### Personal growth

The pandemic also brought with it opportunities for personal growth among participants. This involved increased self-reflection, the opportunity to 'slow down' more outside of work and a perceived improvement in non-work relationships.

### Increased reflection

Participants reported that they were able to reflect more on 'what matters' in their lives. Commonly this was spending quality time with friends and family and appreciating the small things in life.

> one benefit is that actually we quite like a simple life and actually you come to appreciate the very simple things, which are just being outside, going for a bike ride, having a picnic…just the health and happiness of your own family is what's important and everything else, you can generally sort out (Participant_17_GP)

> I think that's one of the positive things that have come out of this whole virus, it that it's allowing people to take a lot more time to reflect on themselves…to reflect on what are the things that really matter, what do I really value in life? (Participant_10_social_worker)

### Slowing down

Participants discussed how the pandemic had given them a chance to slow down and have more 'me-time' and expressed this was something they wished to take forward beyond the pandemic. This view may seem in conflict with the increased workload experienced by certain participants, but reflected the changes outside of work, for example, having fewer social obligations due to social distancing restrictions.

> We have really busy lives generally, and we spend a lot of time rushing around doing lots of stuff, and actually this time has been quite nice in many ways as a period to kind of slow down a bit and I think just appreciating each other. (Participant_1_Hospital_doctor)

> I think it has definitely made me realise that doing less is better for me in terms of not trying to have a finger in every single pie. So I hope I will be able to retain that side of it. (Participant_4_Hospital_ doctor_palliative_care_registrar)

### Improved non-work relationships

While not unanimous, participants talked about how some of their relationships with others improved during the pandemic, especially with family members they lived with, the crisis having brought them closer.

> I've connected with my family a lot more… I feel really good about spending more time with my daughter…that's time that I would never have had with her, so that's really special. (Participant_5_academic_respiratory_physiotherapist)

> As a family, I think we've definitely become closer… we've managed to do some things together which we normally wouldn't do. (Participant_11_Family_support_worker)

## DISCUSSION

This qualitative interview study explored the psychosocial impact of the COVID-19 pandemic on health and social care professionals in the UK. We identified five key themes shared between professionals' accounts. The main difficulties reported were 'communication challenges' (consisting of 'virtual consulting' and 'difficult conversations') and 'work-related stressors' (consisting of 'need to protect loved ones', 'public not following rules' and 'increased workload

and changing work conditions'). Three factors appeared to mitigate some of the psychological distress of the pandemic: 'support structures' (consisting of 'team unity', 'leadership' and 'social support'), 'resilience' (consisting of 'proactive coping', 'accepting uncertainty' and 'increased sense of purpose and reward') and 'personal growth' (consisting of 'slowing down', 'increased reflection' and 'improved relationships').

The themes were drawn from interviews with professionals working in different areas of health and social care, and some themes were felt more strongly in certain jobs than others. For example, GPs and social workers enjoyed the efficiency of new virtual consultations, but GPs found it difficult to use with older adults and those with age-related cognitive decline. Those working with mental health clients, such as psychotherapists and social workers, experienced digital connectivity issues when communicating with vulnerable clients, leading to frustrating repetitions and difficulties building a trusting relationship. This corroborates findings from previous research on barriers to virtual consulting.[29] Those working with older adults, found one of the most challenging elements of the pandemic was having difficult conversations over the telephone with loved ones of patients who were dying—often having to convey that visiting restrictions meant they would be unable to say goodbye in person. This is potentially concerning for the well-being of patients and healthcare professionals, given previous research has highlighted the importance of an appropriate physical and social setting in breaking bad news, and also the presence of family members.[30 31]

There were a number of common work-related stressors among participants. One of these was frustration with members of the public not following the social distancing and hygiene regulations, and with the government's handling of the pandemic. Some also expressed frustration at the 'culture of blame' that they felt permeated the media and public discourse, which can be maladaptive and harmful for one's own mental well-being.[32] Emotional and physical fatigue were also common experiences across all professions, corroborating qualitative studies of health and social care workers during COVID-19 and previous pandemics globally.[4 5 33] Many participants were worried about putting their loved ones at risk of catching COVID-19 as highlighted by previous research,[34] however in our study it appears to have been a particular concern among carers and hospital doctors/nurses, perhaps due to their higher level of virus exposure.

Most participants reported an increased sense of team unity; that they were 'all in this together' fighting a common enemy. This may have led to a degree of resilience against some of the stressful elements of working in a pandemic, supporting findings from previous pandemics.[5 10] Moreover, supporting Berkman's social networks theory[35] which informed part of our interview guide, strong social relationships were frequently cited by participants as key supportive mechanisms for their mental health during this period, including supportive partners at home, friends, family and colleagues. Those participants that did experience loneliness

at work during the pandemic were lone workers, working from home or described unsupportive management. In line with our findings, a recent systematic review of quantitative studies examining the impact of COVID-19 on healthcare workers found social support to be a vital resource underlying their ability to cope.[21]

Participants also described using their own internal resilience as a way to buffer many of the key stressors involved in working through the pandemic, and even thrive at times. They adopted proactive coping mechanisms, as seen in healthcare workers during SARS.[36] These often involved partaking in activities that now have substantial evidence for their role in improving mental health including: exercise,[37] arts and crafts,[38] spending time in nature,[39 40] virtual activities with friends and maintaining a healthy routine.[41 42] Consistent with research from previous pandemics and recent quantitative data during COVID-19, participants also restricted their news intake, particularly as they felt the constant reporting of COVID-19 and the prevalent discourse of blame negatively impacted their mental health.[6 23]

Most participants said they were successfully able to 'let go' of aspects of the pandemic that they felt were out of their control, such as the overall course of the virus and government restrictions. This demonstrates a psychological theory known as 'radical acceptance',[43] and may have been responsible in part for the resilience reported by many participants, also having been identified as a successful coping strategy for healthcare workers in previous pandemics.[36] Some academics have critiqued 'resilience' as a concept for its focus on individual-level rather than structural-level factors,[44] however, participants in the study highlighted the link between these factors, particularly the importance of social networks and social support structures at work. Most participants also expressed gratitude for being able to continue working and described a sense of increased purpose and reward for being able to contribute during the pandemic.

Common 'personal growth' themes were frequently described in participant accounts. Most participants reflected more on 'what matters' in life during this period, which included relationships with friends and family, their health and the health of loved ones and 'appreciating the small things' in life. These findings mirror international qualitative studies looking at the psychological impact of COVID-19 and previous pandemics on health and social care workers which found they experienced 'growth under pressure'[5 33] and increased gratitude and self-reflection.[33] 'Growth under pressure' may be a closely linked (but slightly diluted) concept to 'post-traumatic growth' seen in individuals experiencing personal growth in the aftermath of highly challenging life crises.[45] Lastly, in relation to one of the key theories which informed our interview guide, participants demonstrated a high 'SOC', which may have enhanced their ability to cope during stressful experiences.[19] Participants spoke about 'manageability', whereby they were highly proactive in their coping mechanisms, 'comprehensibility' in their enhanced understanding of the

virus and need for social distancing restrictions, and 'meaningfulness' in how they experienced a heightened sense of purpose through their contribution during the pandemic.

## Strengths and limitations

This is the first known study in the UK to interview both health and social care professionals working in a range of settings on their experiences working through COVID-19, which we felt important as they all continued to provide vital frontline care during the pandemic. This study used a strong theoretical approach to inform the topic guide, and one-to-one interviews allowed in-depth analysis of the psychosocial experiences of health and social care professionals, complementing the wider breadth of quantitative evidence. There were also some limitations. First, we interviewed a wide range of professions, which provided breadth of experience but might limit the specificity of findings. However, due to similarities in the roles of health and social care professionals we felt it important to include a range of voices. Second, given the fluctuating nature of the pandemic, attitudes of health and social care professionals may change over time. This can be difficult to capture during a single interview, however we did ask questions on how their experience had progressed longitudinally. Third, our sample may have been biased towards people who had more free time to participate and so were coping better than others. However, our sample still described a number of stressful experiences during the pandemic, and it is equally possible that workers who were frustrated or stressed wished to express their views. Lastly, interviews were conducted over the phone or video which may have limited the degree to which participants felt able to express themselves, however it may also have been that some participants felt more comfortable communicating this way. Moreover, it was also necessary during the times of the pandemic and also allowed greater uptake, convenience and good regional spread.

## Implications

This study has important implications for health and social care workers, managers, commissioners of services and policy makers during the ongoing pandemic and beyond. First, it highlights the key stressors experienced by health and social care professionals during the COVID-19 pandemic. Many of these echo findings from previous epidemics, but while this is reassuring in terms of data credibility, it highlights a concerning lack of improvement in working conditions during such emergencies over the past two decades. It is vital that the challenges identified here are addressed. Health and social care professionals navigating difficult conversations via telephone or video may benefit from extra training and support at work, for example, in use of the WIRE-SPIKES protocol for breaking bad news remotely.[46] Furthermore, this study provides evidence for the supportive and coping mechanisms used by workers who experienced resilience during this period. Application of coping strategies including leisure activities were common and reportedly beneficial, as were the use of mindful techniques such as expressing gratitude. This suggests that

health and social care professionals may benefit from regular work-based interventions providing space for such activities. While such activities may feel extraneous during emergency situations, the building of resilience and positive coping outside of pandemic situations and the tackling of problems such as staff burnout will likely improve staff coping capacity in future epidemic situations. Alongside this, adequate provision for social support should be ensured, from family and friends and via the work place, for example, through enhanced supervision or peer support. The research presented here suggests that investment into well-being support could play a vital role in helping health and social care workers to manage emotional stress.

## CONCLUSION

To the best of our knowledge, this is the first qualitative study to explore the psychosocial impact of the COVID-19 pandemic on both health and social care professionals working in different settings across the whole of the UK. Participants experienced communication challenges and changing work conditions, but also positive factors such as increased team unity, and greater reflection on what matters in their life. This study offers important evidence for continued and future disruptions caused by the COVID-19 pandemic. It also elucidates successful psychological and practical strategies deployed by health and social care professionals that could be used to support their resilience and well-being.

**Acknowledgements** The authors would like to thank other members of the qualitative COVID-19 social study team at UCL, who helped discuss themes during the analysis stages at weekly team meetings: Dr Thomas May, Dr Anna Roberts and Dr Joanna Dawes.

**Contributors** DF, AB and HA conceived the study and contributed to the study design. HA conducted all the interviews apart from 1, which AB conducted. HA coded all the transcriptions. ARMcK coded four transcripts for cross-checking purposes. HA wrote up the manuscript. All authors critically reviewed the manuscript and approved the final submission.

**Funding** This COVID-19 Social Study was funded by the Nuffield Foundation (WEL/FR-000022583), but the views expressed here are those of the authors. The study was also supported by the MARCH Mental Health Network funded by the Cross-Disciplinary Mental Health Network Plus initiative supported by UK Research and Innovation (ES/S002588/1), and by the Wellcome Trust (221400/Z/20/Z). DF was funded by the Wellcome Trust (205407/Z/16/Z).

**Competing interests** None declared.

**Patient consent for publication** Not required.

**Ethics approval** The study was reviewed and approved by the UCL Ethics Committee (Project ID 14895/005).

**Provenance and peer review** Not commissioned; externally peer reviewed.

**Data availability statement** All data relevant to the study are included in the article or uploaded as supplementary information. Raw data are available in the form of participant quotations and the interview topic guide.

terminology, drug names and drug dosages), and is not responsible for any error and/or omissions arising from translation and adaptation or otherwise.

**ORCID iDs**
Henry Aughterson http://orcid.org/0000-0001-5568-6474
Daisy Fancourt http://orcid.org/0000-0002-6952-334X

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
