## [Reviewer comments · BMJ Open]

ARTICLE DETAILS

TITLE (PROVISIONAL)	The psychosocial impact on frontline health and social care professionals in the UK during the COVID-19 pandemic: a qualitative interview study.
AUTHORS	Aughterson, Henry; McKinlay, Alison; Fancourt, Daisy; Burton, Alexandra

VERSION 1 – REVIEW

REVIEWER	Annette Peart Monash University, Australia
REVIEW RETURNED	03-Dec-2020

GENERAL COMMENTS	This paper is concisely written and clear to read. It is obviously an important paper while UK is still in the midst of the pandemic. However there are some design issues that need to be addressed. 1. There does not appear to be a clear rationale for the focus on resilience (the second aim of the study).2. Please outline in your methods how you have ensured the reliability and validity of this research (also referred to as trustworthiness). A good paper to review these concepts is available here https://ebn.bmj.com/content/18/2/34. Without a discussion of credibility and transparency of your research process, it is difficult to say whether your results did address your aims.3. When writing up the results, please keep in mind that participants are REPORTING experiencing certain feelings - you have not observed these directly, so it is technically not correct to say "participants experienced", in fact "participants reported experiencing ...".4. Please review aspects of the results where you have quantified how many participants said a particular thing. 'Some' or 'most' are irrelevant to the write up of the qualitative data. What is most important is WHAT is said, not how many said it.5. However, in saying that, presenting just one piece of data to support a theme is inadequate. This may reflect a lack of description of the research process. It is hard to see where the data analysis occurred, as this reads as a description of the data that 'matched' the themes. Data related to 'need to protect loved ones', 'public not following the rules', 'uncertainty of risk to patients', and under 'personal growth' are single cases, which do not make a 'theme', unless there is evidence of a deeper process of analysis.6. It may be useful to go back to the analysis and see where some sub-themes can be integrated into a theme. Themes do not have to have sub-themes. With so many sub-themes, I got lost in what you were actually trying to say.
--

	7. While you refer to theory in the introduction, and that it helped inform your interview questions, there is not really any evidence to confirm this. Perhaps this needs to be made more explicit. The theory was not adequately returned to in the discussion to be able to say this project was informed by theory. 8. Where you state 'this is the first study ...' - it actually is the first *known* study - there are others out there, but they may not be known to you. I would be happy to read a revised version of this paper. Once you have clearly communicated your design and processes, it will be an important contribution.
--	--

REVIEWER	Van Royen Paul University of Antwerp, Faculty of Medicine and Health Sciences, department of Family Medicine and Population Health
REVIEW RETURNED	24-Dec-2020

GENERAL COMMENTS	This manuscript is well written and deals with a very actual topic- so certainly a paper 'of the moment'. I have some minor remarks mainly on the used methodology and reporting of this:  - Since no grounded theory design or methodology was used, data saturation (page 4 line 53) is not the right term to be used. It is better to mention that data sufficiency (with enough richness and depth of data) was reached - It is useful to give more details on the used analysis method- especially on the different steps of thematic analysis- for instance how exactly was searched for themes by using tables of mind-maps. How were the themes reviewed - it is only mentioned that first a deductive approach was used followed by an inductive one. More explanation on this could clarify and underpin the audit trail - Should the fact that interviews were conducted over the phone or video call, not be included in the limitations? How did this decreased the in depth inquiry of the psychosocial impact ?
--

VERSION 1 – AUTHOR RESPONSE

Responses (in asterisks) to reviewers' comments

Reviewer: 1

Responses to reviewer comments

This paper is concisely written and clear to read. It is obviously an important paper while UK is still in the midst of the pandemic. However there are some design issues that need to be addressed.

1. There does not appear to be a clear rationale for the focus on resilience (the second aim of the study).

*** We felt it was important to explore both the 'psychosocial impact' on health and care workers, but also those factors which either 'alleviated distress'/supported mental health, or contributed to their resilience – since resilience is a concept linked to one's ability to effectively cope with stressors e.g. those brought about by the pandemic.

Antonovsky's sense of coherence theory, which we lay out in the third last paragraph of the introduction, is a theory of resilience. There may be some commonly shared factors that contribute to

workers' ability to cope with stressors brought about by the pandemic, and these factors may be better harnessed by employers or individuals. So, the second aim of the study is to explore both these factors which may:

- i) alleviate distress/support well-being, and
- ii) contribute to resilience.

There is a slight distinction between i) and ii) in that i) may involve activities such as hobbies and ii) may include more psychological strategies such as 'accepting uncertainty', however there is of course significant overlap between factors which 'alleviate distress/support well-being' and 'contribute towards resilience'.

We have now made these points clearer in the final and third last paragraph of the introduction. (pg3)

2. Please outline in your methods how you have ensured the reliability and validity of this research (also referred to as trustworthiness). A good paper to review these concepts is available here <https://ebn.bmj.com/content/18/2/34>. Without a discussion of credibility and transparency of your research process, it is difficult to say whether your results did address your aims.

*** Thank you for the link to this highly useful paper. We have now made changes in the methods section, in order to demonstrate how we enhanced the trustworthiness of our results. In particular, we have better explained how we reduced individual bias, e.g. using weekly peer meetings.

in order to stay true to the original accounts we conducted repeated revisiting of the audio-recorded transcripts. Additionally, we have now explained how we provided rich and verbatim descriptions of participants' accounts in order to support our findings.

Our strength and limitations section also reflects on the potential sample bias. Moreover, in the strength and limitations section we reflect on how the wide range of participants we interviewed might limit 'specificity' of results, but also might help provide a good breadth of experience across different settings – relating to 'applicability' of results as described in the attached paper.

We have also reported our methods in line with the COREQ criteria, which includes providing information on reflexivity, our research team, study design and data analysis and so this also helps ensure trustworthiness. (pg4-5) ***

3. When writing up the results, please keep in mind that participants are REPORTING experiencing certain feelings - you have not observed these directly, so it is technically not correct to say "participants experienced", in fact "participants reported experiencing ...".

*** Thank you for this important clarification. Throughout the manuscript, where we have just written 'experienced' we have replaced with 'reported they experienced' or 'reported' or 'said'. ***

4. Please review aspects of the results where you have quantified how many participants said a particular thing. 'Some' or 'most' are irrelevant to the write up of the qualitative data. What is most important is WHAT is said, not how many said it.

*** Again, thank you for this clarification. We have now amended throughout the manuscript to remove the 'most' and 'some' and 'many's etc.

There are a very small number of occasions where we have left in a quantifying adjective e.g. 'some', if it was deemed necessary in order not to present seemingly contradictory data that might confuse

the reader – e.g. some participants felt they had more time to ‘slow down’ in other areas of life despite the increased workload/working hours (which might otherwise seem contradictory). ***

5. However, in saying that, presenting just one piece of data to support a theme is inadequate. This may reflect a lack of description of the research process. It is hard to see where the data analysis occurred, as this reads as a description of the data that ‘matched’ the themes. Data related to ‘need to protect loved ones’, ‘public not following the rules’, ‘uncertainty of risk to patients’, and under ‘personal growth’ are single cases, which do not make a ‘theme’, unless there is evidence of a deeper process of analysis.

*** We have now included additional evidence (in the form of further quotes and written analysis) to support these sub-themes.

Quotes had been purposively limited beforehand in an attempt to not over-load the reader/reduce word count. These sub-themes did all have significant depth behind them, and were not just single cases but represented to us clear patterns in the data. The one exception to this, on closer examination, appears to be ‘uncertainty of risk to patients’ – we have now removed this as a sub-theme, and merged this with ‘increased workload and changing working conditions’. ***

6. It may be useful to go back to the analysis and see where some sub-themes can be integrated into a theme. Themes do not have to have sub-themes. With so many sub-themes, I got lost in what you were actually trying to say.

*** Thank you for this comment - we did a significant amount of work cutting down our sub-themes prior to submission – but do feel, with only 5 broader themes, it is possible for us to present our data more clearly. Each sub-theme, in our view, does represent a rich pattern of discreet data. We have also now added in a signposting/explanatory sentence at the start of each theme to help navigate the reader so the sub-themes/themes are clearer to follow, and more obviously linked with each theme. We feel this helps the results section read much more clearly. ***

7. While you refer to theory in the introduction, and that it helped inform your interview questions, there is not really any evidence to confirm this. Perhaps this needs to be made more explicit. The theory was not adequately returned to in the discussion to be able to say this project was informed by theory.

***In the data collection section where we mention the theories which informed the topic guide questions, we have now given examples of the questions we asked relating to the theories. (pg4)

In the discussion section we discuss the social networks theory and how our findings supported this. In the final paragraph of the discussion we have discussed the ‘sense of coherence’ and how participants demonstrated they had a fairly strong ‘sense of coherence’. We explain how each component of this theory (manageability, meaningfulness and comprehensibility) was demonstrated in our findings. It was perhaps not clear enough at the end that this referred to one of the theories which informed our interview guide and we have now made this more explicit. (pg3,4 and pg15,16) ***

8. Where you state ‘this is the first study ...’ - it actually is the first *known* study - there are others out there, but they may not be known to you.

*** Thank you – we have now replaced ‘first study’ with ‘first known study’. (pg1 and pg16) ***

I would be happy to read a revised version of this paper. Once you have clearly communicated your design and processes, it will be an important contribution.

Reviewer: 2

Responses to reviewer comments

This manuscript is well written and deals with a very actual topic- so certainly a paper 'of the moment'. I have some minor remarks mainly on the used methodology and reporting of this:

- Since no grounded theory design or methodology was used, data saturation (page 4 line 53) is not the right term to be used. It is better to mention that data sufficiency (with enough richness and depth of data) was reached

*** Thank you for this clarification - we have now replaced the term 'data saturation' with 'data sufficiency' when it has been used. (pg4-5) ***

- It is useful to give more details on the used analysis method- especially on the different steps of thematic analysis- for instance how exactly was searched for themes by using tables of mind-maps. How were the themes reviewed - it is only mentioned that first a deductive approach was used followed by an inductive one. More explanation on this could clarify and underpin the audit trail

*** We have given more information on our thematic analysis methods, e.g. providing a clearer explanation of our inductive approach, and also provided more information on how we reduced bias during data analysis. (pg5) ***

- Should the fact that interviews were conducted over the phone or video call, not be included in the limitations? How did this decreased the in depth inquiry of the psychosocial impact ?

*** We have now included this in our limitations section at the end, making it clear this might be one limitation, but also that some participants may have actually been more comfortable expressing themselves this way, and also that it was absolutely necessary during the times of the pandemic, and helped aid convenience for participants and good regional spread. (pg16-17) ***

VERSION 2 – REVIEW

REVIEWER	Annette Peart Monash University, Australia
REVIEW RETURNED	19-Jan-2021

GENERAL COMMENTS	Thank you for providing additional context and details for this paper. The additions are clear and concise. Well done.
--

REVIEWER	Van Royen Paul Department of Family Medicine and Population Health, Faculty of Medicine and Health Sciences, University of Antwerp, Belgium
REVIEW RETURNED	28-Jan-2021

GENERAL COMMENTS	The authors have well addressed all the given comments.
---